# Faster Neural Networks Straight from JPEG

**Lionel Gueguen & Alex Sergeev**
Uber Technologies
San Francisco, CA 94103, USA
{lgueguen, asergeev}@uber.com

**Rosanne Liu & Jason Yosinski**
Uber AI Labs
San Francisco, CA 94103, USA
{rosanne, yosinski}@uber.com

## Abstract

Training CNNs directly from RGB pixels has enjoyed overwhelming empirical success. But can more performance be squeezed out of networks by using different input representations? In this paper we propose and explore a simple idea: train CNNs directly on the blockwise discrete cosine transform (DCT) coefficients computed and available in the middle of the JPEG codec. We modify libjpeg to produce DCT coefficients directly, modify a ResNet-50 network to accommodate the differently sized and strided input, and evaluate performance on ImageNet. We find networks that are both faster and more accurate, as well as networks with about the same accuracy but 1.77x faster than ResNet-50.

## 1 Introduction

Progresses toward training convolutional neural networks on a variety of tasks (Krizhevsky et al., 2012; Mnih et al., 2013; Ren et al., 2015; He et al., 2015) has led to the widespread adoption of such models in both academia and industry. Traditionally CNNs are trained with input provided as an array of red-green-blue (RGB) pixels. In this paper we propose and explore a simple idea for accelerating neural network training and inference where networks are applied to images encoded in the JPEG format. We modify the libjpeg library to decode JPEG images only partially, resulting in an image representation consisting of a triple of tensors containing discrete cosine transform (DCT) coefficients in the YCbCr color space. Due to how the JPEG codec works, these tensors are at different spatial resolutions. We then design and train a network to operate directly from this representation; as one might suspect, this turns out to work reasonably well. Fig. 1ab shows the JPEG encoding process and a schematic view of the partial decoding process we employ in this paper. JPEG encoding consists of three steps: first the color space of an image is converted from RGB to YCbCr, the spatial resolution of the latter two channels is often reduced. Then each of the three channels is split into blocks of $8 \times 8$, and each block undergoes a DCT. Lastly the result of DCT goes into a Huffman encoding algorithm for further compression . In our work we decode a compressed image up to its DCT coefficients, which are then directly input to a CNN.

## 2 Designing CNN models for DCT input

We adapt ResNet-50 (He et al., 2015) for processing DCT coefficients. Some care is required, as DCT coefficients from the Y (*luma*) channel, $D_Y$, generally have a larger size than those from the *chroma* channels, $D_{Cb}$ and $D_{Cr}$, as shown in Fig. 1a. We design two special transforms $(T_1, T_2)$ that take care of the spatial dimension matching. Fig. 1c illustrates this process. We consider the receptive field size and stride (hereafter denoted with $\mathcal{R}$ and $\mathcal{S}$) for each unit at the end of transforms and throughout the network. Whereas for typical networks taking RGB input, the receptive field and stride of each unit will be the same in terms of each input channel (red, green, blue), here the receptive fields considered *in the original pixel space* may be different for information flowing through the Y channel vs the Cb and Cr channels, which may not be desired. We explored seven different methods of transforms $(T_1, T_2)$, from the simplest upsampling to deconvolution, and combined with different options of subsequent ResNet block stacks. **UpSampling:** Both *chroma* DCT coefficients $D_{Cb}$ and $D_{Cr}$ are upsampled by a factor of two in height and width to the dimensions of $D_Y$. The three are then concatenated, and go through a batch normalization layer before going into ResNet blocks 3, 4, and 5. **UpSampling-RFA:** This setup is similar to UpSampling, but here we keep ResNet Block 2 (rather than removing it), and modify blocks 2 and 3 such that they mimic the increase in $\mathcal{R}$ and $\mathcal{S}$ observed in original ResNet-50; we denote this "Receptive Field Aware" or RFA. **Deconvolution-RFA:** An alternative to upsampling is a learnable deconvolution layer. In this design we use two

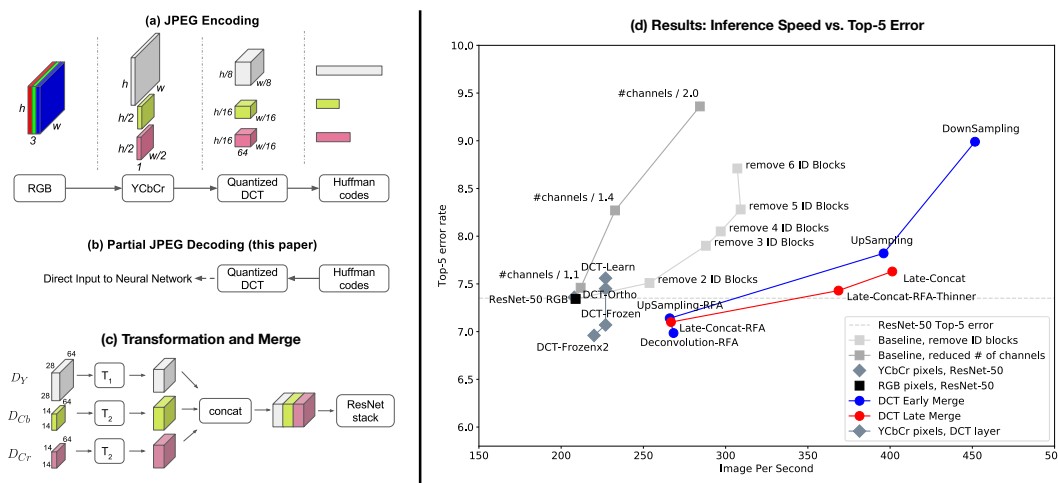

Figure 1: (a) The three steps to encode JPEG images. (b) The inverse process of JPEG decoding. In this paper we run only the first step of decoding and then feed directly into a neural network. (c) The coefficients $D_Y$ and $D_{Cb}, D_{Cr}$ are transformed through $(T_1, T_2)$ into activations of identical spatial sizes. (d) Inference speed vs top-5 error rates. Six group of experiments are presented. ResNet-50 baseline on both RGB and YCbCr show nearly identical performance, indicating that the YCbCr color space on its own is not sufficient for improved performance. Two sets of RGB controls show that simply making ResNet-50 shorter or thinner cannot produce competitive speed gains. Finally, two sets of DCT experiments are shown, those that merge Y and Cb/Cr channels early in the network (within one layer of each other) or late. Several of these networks are both faster and more accurate, and the Late-Concat-RFA-Thinner network is about the same level of accuracy while being 1.77x faster than ResNet-50.

separate deconvolution layers on $D_{Cb}$ and $D_{Cr}$ to increase of spatial size. Channel size is kept the same. The rest of the design is the same as UpSampling-RFA. **DownSampling:** As opposed to upsampling spatially smaller coefficients, another approach is to downsample the large one, $D_Y$, with a convolution layer. The rest of the design is the same as UpSampling. As we will see in Sec. 3, this network operates on smaller total input, resulting in much faster processing at the expense of higher error. **Late-Concat:** In this design, we run $D_Y$ through Block 3 of ResNet-50 given that they share the same spatial dimensions. In parallel, $D_{Cb}$ and $D_{Cr}$ are passed through an ID block, before being concatenated with the $D_Y$ path. The joined representation is then fed into Blocks 4 and 5. This results in more total computation along the luma path than the chroma path and tended to result in fast networks with good performance. **Late-Concat-RFA:** This receptive field aware version of Late-Concat passes $D_Y$ through modified Block 2 and 3, such that the increase in $\mathcal{R}$ mimics the $\mathcal{R}$ in the original ResNet-50. Then, $D_{Cb}$ and $D_{Cr}$ are concatenated to the activations of the first layer from Block 4. Because a smaller spatial size is used in Block 2, we increase the number of channels to 1024 to keep the representation size at that layer the same. **Late-Concat-RFA-Thinner:** This architecture resembles Late-Concat-RFA where the number of channels is decreased in Block 2 and 3. Instead of letting the number of channels rise to 1024 in Block 2, and then 512 in Block 3, the number of channels is set to 384 in both blocks. Then, the Cb and Cr components with 64 channels each, are convolved into 256 channels while maintaining their spatial dimensions. These activations get concatenated with the activations from the first identity layer from the block 4, allowing to form a volume of size $14 \times 14 \times 1024$. These changes were tried in an attempt to keep the performance of the Late-Concat-RFA model but obtain some of the speed benefits of the Late-Concat. As will be shown in Fig. 1d, it strikes an attractive balance.

# 3 RESULTS AND DISCUSSIONS

We train on ImageNet (Deng et al., 2009) with the standard ResNet-50 stepwise decreasing learning rates described in (He et al., 2015). The distributed training framework Horovod (Sergeev & Balso, 2017) is employed to facilitate parallel training over 128 GPUs. Experiments are conducted with images which are first resized to 224×224 pixels with a random crop, and the JPEG quality used during encoding is 100%, so as little information is lost as possible.

**ERROR RATE VS INFERENCE SPEED —** Results are shown in Fig. 1d for all seven proposed DCT architectures from Section 2, along with two baselines: ResNet-50 on RGB inputs, and ResNet-50 on YCbCr inputs. The full result include validation top-1 and top-5 error rates and inference frames per second (FPS). Both ResNet baselines achieve a top-5 error rate of 7.35% at an inference speed of 208 FPS on a Titan GPU, while the best DCT network achieves it at 6.98% with 268 FPS. We further analyze the results by dividing them into three categories, and make our conclusions in each.

**UpSampling, DownSampling, Late-Concat**. In these architectures DCT coefficients are directly connected to ResNet-50. Several of these architectures providing significant inference speed-up (three far-right dots in Fig. 1d), almost $\times 2$ in the best case. The speedup is due to less computation as a consequence of reduced ResNet blocks. A sharp increase of error with DownSampling suggests that a reduction in the spatial structure of the Y (*luma*) causes a reduction of information, while maintaining its spatial resolution (as in UpSampling and Late-Concat) performs closer to the baseline.

**UpSampling-RFA, Late-Concat-RFA** the two best architectures above are extended to slowly increase their $\mathcal{R}$, so as to mimic the $\mathcal{R}$ growth of ResNet-50. They are shown to achieve better error rates than their non-RFA counterparts while still providing an average speed-up of $\times 1.3$. With the proper RFA adjustments in architecture, these two versions manage to beat the RGB baseline.

**Deconvolution-RFA, Late-Concat-RFA-Thinner** In this category we attempt to further improve the RFA architectures, by (1) learning the upsampling operation with Deconvolution-RFA, and (2) reducing the number of channels with Late-Concat-RFA-Thinner. On the one hand Deconvolution-RFA reduces the top-5 error rate of UpSampling-RFA by $0.15$ while maintaining an equivalent inference speed. On the other hand, Late-Concat-RFA-Thinner achieves error rates on par with the baseline while providing a speed-up ratio of $\times 1.77$.

**RGB NETWORK CONTROLS —** For a fair comparison, we make control networks with architecture tweaks similar to what we did with our proposed DCT networks. We mutate the ResNet architecture slightly to get it to perform with lower error and/or higher speed. The experiments and observations are summarized below.

**Reducing the Number of Layers**. We remove the Identity blocks one at a time, from the bottom up, from Blocks 2 and 3, resulting in 6 experiments as 6 layers are removed. We can see the trade-off between the inference speed and accuracy in Fig. 1d under the legend "Baseline, Remove ID Blocks" (series of 6 gray squares). As can be seen, networks become slightly faster but at a large reduction in accuracy.

**Reducing the Number of Channels**. We also investigate thinning the network to speed up inference. We reduce layer width by each of three ratios: $\{1.1, \sqrt{2}, 2\}$. The same trade-off between speed and accuracy is shown in Fig. 1d under the legend "Reduced # of Channels". As with reducing the number of layers, networks become slightly faster but at a large reduction in accuracy.

**Learning the DCT Transform**. A final set of four experiments — shown in Fig. 1d as four "YCbCr pixels, DCT layer" diamonds — addresses whether we can obtain similar benefit to the DCT architectures but starting from RGB pixels by processing RGB pixels using convolutional layers designed to replicate, exactly or approximately, the DCT transform. We use $8 \times 8$ filters with stride 8. In DCT-Learn, we randomly initialize filters and train them in the standard way. In DCT-Ortho, we regularize the convolution weights toward orthonormality, as described in Brock et al. (2017), inspired by the orthonormality of the DCT transform. In DCT-Frozen, we use the exact DCT coefficients without training, and in DCT-Frozenx2 we modify the stride to be 4 instead of 8 to increase representation size at that layer and allow filters to overlap. Surprisingly, this network tied the performance (6.98%) of the best other approach when averaged over three runs, though without the speedup of the Deconvolution-RFA approach. This is interesting because it departs from network design rules of thumb currently in vogue: first layer filters are large instead of small, hard-coded instead of learned, run on YCbCr space instead of RGB, and process channels depthwise (separately) instead of together. Future work could perhaps evaluate to what extent we consider adopting some of these choices as standard practice.

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
