# OpenReview forum: "Faster Neural Networks Straight from JPEG"
_ICLR.cc/2018/Workshop — Accept_

### Official Review · AnonReviewer2 · 2018-02-27
**An exhaustive study with an unexpected conclusion**

**Rating:** 7
**Confidence:** 4

**Review:**

This work explores the behavior of ResNet-50 for ImageNet classification when applied directly over the DCT coefficients of JPEG images. The authors present seven different variations that obtain both gains in accuracy and speed. The experiments also include a configuration in which the first layer mimics the actual DCT filters, obtaining the same gain in accuracy as the best learned architecture.

The presented work is exhaustive and original in terms of how to explore the potential of working over DCT coefficients. the authors consider both the accuracy and speed dimensions on the results, and actually obtain interesting conclusions that may impact real-world products that may value the trade off between them. The text is clear, well-written, and guides the reader into the understanding of the figures and results.

PROS
- The different considered configurations are deeply discussed and argued.
- The gains obtained in the DCT-Frozen configuration might have an impact in other tasks. These results open the door to further exciting research.
- Text is clear and figures informative.

CONS
- Given some surprising results, it would be advisable to run the experiments more than 3 runs and study the variance of the results.
- The results on Top-1 error rate are not included, and it is not discussed either if they match the conclusions obtained over Top5 error rate.

- The paper fails into citing the related work in the field of image understanding from the compressed domain, which obviously limits the novelty of the work. For example:

Fu, D., & Guimaraes, G. (2016). Using Compression to Speed Up Image Classification in Artificial Neural Networks.

Javed, M., Nagabhushan, P., & Chaudhuri, B. B. (2017). A review on document image analysis techniques directly in the compressed domain. Artificial Intelligence Review, 1-30.

Robert Torfason, Fabian Mentzer, Eirikur Agustsson, Michael Tschannen, Radu Timofte, Luc Van Gool
Towards Image Understanding from Deep Compression Without Decoding (ICLR 2018)
https://openreview.net/forum?id=HkXWCMbRW

---

### Official Review · AnonReviewer1 · 2018-03-09
**nice demonstration of effective input representation**

**Rating:** 7
**Confidence:** 3

**Review:**

This paper demonstrates the use of DCT coefficients available in the JPEG image format as an effective input representation to a convolutional network.  Instead of inputting RGB pixels, a JPEG-compressed image is half-decompressed to its 8x8 block DCT coefficients, which are used as input to the network.  8x8 strides are similar to existing ResNet-50 layer 2 spatial resolution, and several approaches are evaluated for how to sample and place the DCT input into the network.  Evaluations are performed on ImageNet classification; the method can achieve good performance while offering significant speedups, mostly by being able to replace the first two blocks with already-computed JPEG DCT.

This is an interesting and effective technique, and I found the paper good overall.  I think it can be improved in much of its description of the different architecture variants --- these were hard to follow from the descriptions alone, and I think could much benefit from a picture of these.

Also I think additional discussion on related and prior work might be good.  One recent paper I was able to find pretty quickly is Ulicny and Dahyot IMVIP 2017 "On using CNN with DCT based Image Data".

Another experiment I would be interested in is how does JPEG quality affect the results?  Can the network be trained with DCT inputs from different quality/compression settings (or a range of quality)?  But I don't think that is necessary for the workshops.

Overall, this is a nice demonstration, and while I think it could be a little clearer in the exact model modifications, the idea is quite nice to see and effective.

---

### Official Review · AnonReviewer3 · 2018-03-09
**A good idea, and correct benchmark to demonstrate efficient results, but short discussion**

**Rating:** 7
**Confidence:** 3

**Review:**

This paper deals with image representation, and use of the compressed representation of JPEG to accelerate recognition.
Performance on ImageNet are interesting.

The discussion is, due to the format, quite short.
Shall be extended, reducing even more other parts.

For example we would like to have idea of the impact of the JPEG quality Q
used during encoding.
Here it is 100%, but what is the compromise Q, speed and accuracy ?

---

### Decision · Program_Chairs · 2018-03-20
**ICLR 2018 Workshop Acceptance Decision**

**Decision:**

Accept

**Comment:**

Congratulations, your paper was accepted to the ICLR workshop.